# Research on the Development of Equitable Education in China from the Human Capability Perspective

**Mingmei Li [†], Min Liu [†], Hejia Wang, Xiaohan Hong and Chen Wang ***

Faculty of Education, Beijing Normal University, Beijing 100875, China
* Correspondence: wangchen@bnu.edu.cn
[†] These authors contributed equally to this work.

**Abstract:** "Capability" is an important conceptual tool for addressing educational inequity (EI). This paper analyzes the existing limitations of developing educational equity in China from the human capital perspective and proposes the human capability approach as a way to improve it. This paper begins by a policy review on China's education equity measures, revealing a troubling emphasis on resources allocation and a "top-down" governance. In response, we propose an actionable research approach as a means to improve multi-stakeholder collaboration in educational equity reform and to further the development of student capabilities. The study also presents a case study to illustrate the process of using "capability" and actionable research methods to promote educational equity, demonstrating the necessity and effectiveness. We also note that education inequality is a delicate and complicated topic that requires joint, flexible and innovative efforts.

**Keywords:** capability; equitable inequality (EI); equitable education (EE); actionable research; LFCF

Under the influence of Knowledge Economy, the pursuit of economic values of education not only solidifies the neglect of human subjectivity in educational reforms, but also aggravates educational inequality. China's recent mainstream approach to equitable education (EE) development in poverty-stricken areas is to "empower" the poor through education, making them talented workers, and in return promote local economic development. For example, the difference in education level between regions is often regarded as one of the most important manifestations of the income gap between urban and rural areas in China [1]. In addition, it is widely accepted that promoting balanced development in education between regions is going to help narrow the income gap between regions [2]. However, we believe that education should not regard people as economic tools, and human capability development is the fundamental purpose of education. Hence, education policies guided by human capital theory play a limited role in solving educational inequality. Therefore, we need to complement a theory of education that has human capability development at its core. How should this theory be applied in practice to improve educational inequity? Considering actionable research conforms to the norms of both qualitative and quantitative research and seeks to identify unequal phenomena and their causes in society and education, we try to combine the human capability approach with actionable research, developing new theoretical perspectives and applied frameworks.

## 1. Human Capability Approach and Education Development

As Yates summarized, the transition from human capital theory to human capability development theory will achieve an overall change in educational goals and assessment, thereby driving the innovation of educational practice [3]. Human-capital-oriented education regards development as "growth", pays attention to student learning achievements, and emphasizes the "input–output" evaluation system. Hence, the ratio of educational input to educational output is the key to determining the success of education practices. In this process, students themselves are invisible. Unlike the human capital approach, if education focuses on the development of individual capabilities, then the goal of education

is to promote individual freedom, and the initiative shown by individuals becomes a key factor in evaluating the success of education practices (see Table 1).

**Table 1.** Development discourses and quality learning.

| Discourse: Development As | Notion of Quality Education | Focus of Learning | Evaluative Focus | Agency–Structure Relationship | Underlying Political Philosophy |
|---|---|---|---|---|---|
| Growth—Human capital | Behaviorism | Consequences | Input–output | Intervention | Utilitarianism |
| Improved human rights | Humanism | Constructions | Processes | Institution | Liberalism |
| Liberation | Criticality | Connections | Outputs/outcomes | Interaction | Post-Marxism |
| Enhanced freedoms | Capability | Combinations (3C) "rich learning" | Agency | Integrative (3I) | Globalism Thick cosmopolitanism |

In the above Table, 3C represents a richer form of learning, which regards learning as a consequence, construction and connection (3C); 3I means that this work needs short-term intervention supplemented by long-term institutionalization and sustained social interaction [3] (p. 3).

The capability approach is a theoretical framework that essentially highlights the neglected factors under human capital theory and reshapes the way education is reformed. And it has been widely used to study social inequality in education. According to this theory, people's development should be a process in which individuals regain their subjectivity, gain their capability and realize their cherished life. The application of this conceptual tool can help break the limitation of the human-capital-oriented approach and pursue the development of EE centered on developing people's capabilities.

Sen defines capability as "an optional combination of things that a person can do or can be, that is, various functions that he or she can achieve" [4] (p. 30). Opposing the view of traditional welfare economics that welfare is equal to utility, he thinks that there are two indispensable stages from welfare to utility: capability and function. Functions are realized results such as reading, while capability is the potential to realize these functions such as being taught to read and having books or newspapers to read. Therefore, the difference between capability and function is the difference between realized opportunity and actual achievement, or the difference between potential and result [5] (p. 4).

Sen further distinguished capability from function as follows [6] (pp. 34–35):

- Functioning refers to a person's achievements and what an individual tries to do or be. It reflects, as it were, a part of the "state" of that person;
- Capability refers to a person's ability to achieve a given functioning ("doing" or "being");
- Functioning n-tuple describes the combination of "doings" and "beings" that constitute an individual's life state, with each functional n-tuple representing a possible lifestyle;
- The capability set describes a set of attainable functioning n-tuples that an individual can realize, where an individual can choose between different commodity bundles and utilization methods.

Therefore, a person's capability refers to an alternative combination of functions that they can realize. In other words, they have effective opportunities to engage in their voluntary actions and activities, to have freedom to realize various lifestyles and to become the person they want to be. The capability approach proposed by Sen has been widely accepted and applied because it considers human beings to be the goal, recognizes human heterogeneity and diversity, pays attention to group differences, accepts people's initiative and participation, and recognizes that different people, cultures and societies may have different values and aspirations [6] (p. 34).

The capability approach has important implications for social justice in education. First, education itself is a basic ability, which affects the development and expansion of other abilities [5] (p. 8). This means that if there is an absence or a lack of educational opportunities, essential harm and disadvantage is caused to individuals. Second, educational capability plays a substantial role in expanding other existing and future capabilities, so it is the basis of different capabilities and the possibility of living a better life. Third, the instrumentality and the intrinsic value of education itself can improve individual freedom, including freedom of well-being and freedom of initiative, which are emphasized by the capability approach [7] (pp. 30–31). Finally, in the educational application, human capability provides a set of conceptual tools to think about how to reduce injustices in the current education system and the wider society. As Hart (2012) explains, this set of tools allows us to think creatively about the role, process, and content of education, broadening our horizons beyond the limitations of standardized testing, neoliberal discourse, and quantitative policy directives [8] (p. 278). In this way, we can transcend the limitations of human capital, truly think about the goals and values of education from the perspective of human freedom and human ability development itself, and improve educational inequality.

It is on this basis that many scholars prefer the concept of "educational capability". It refers to offering students with low socio-economic status (SES) rights and information so that they can choose the educational path they value. Cliona pointed out that education reform should aim at expanding human capability and providing activities for students with a low SES to help them fill the gaps in their social and cultural capital [9] (p. 70), since such gaps determine the size of their "capability set" to some extent. Furthermore, education provides opportunities for individuals to transform their capabilities into functions, which is particularly critical to the educational problems faced by vulnerable groups in many low-income countries [10] (p. 395). This means that improving the education of vulnerable groups actually requires us to pay attention to whether the education received by the disadvantaged groups can improve their capability, whether the ability of the disadvantaged group has been freely developed, and whether they can choose the life they want to live with the ability acquired. In other words, equity in education is ultimately a kind of equity in competence.

Although human capability is an important supplementary framework for the research and formulation of the EE policy in China, it is still lacking in partitionable grounds. Most scholars regard it as a theoretical tool to analyze and discuss poverty governance and social welfare issues, and lack the awareness of using action research concepts to solve local problems, failing to form the theoretical connotation and the practical system of feasible ability with local vitality.

Research shows that human capability theory can be an effective tool to promote equitable education development. According to a nationwide empirical study on education satisfaction, people are paying more attention to capability equity than resource equity when it comes to education distribution, and are more concerned about the equity within the organization and the equity that is more closely related to their current experience than the equity in distribution and the equity between organizations, such as narrowing the gap between schools and integrating urban and rural education [11] (p. 39). Meanwhile, capability theory pays more attention to the EE of specific individuals. The realization of such equity depends not only on educational resources and results, but also on the expansion of students' capability or the enhancement of students' initiative through education.

Therefore, based on the current research of human capability theory and troubled practices, we believe that in order to better employ the human capability theory to solve the problem of equitable development of education in the Chinese context, we should further combine actionable research approach to explore feasible ideas of capability expansion, achieve the social welfare needs of the target group, provide a basis for the introduction of relevant social welfare policies, and help the sustainability of China's social welfare needs. On this basis, we also need to develop appropriate solutions to improve educational equality through actionable research to explore feasible capability expansion at the educational

level, both theory-wise and practice-wise; this is the core topic of this research. We aim to promote the development of educational equity through actionable research to expand the viable capabilities of individuals and call for more research investment.

## 2. Equity of Educational Opportunities in China from the Human Capability Approach

As an important social issue in China, EE is commonly associated with topics like economic growth, social solidarity, social equality and justice, and draws wide attention. Since the Tenth National Five-Year Plan for Education Development released in 2001 proposed "EE" as a basic education policy for the first time, China has made documented progress in promoting EE. However, due to China's huge population, social stratification and vastness in territory, the development of EE in China is arduous. Although this kind of extensive resource redistribution among different regions and population groups is conducive to achieving relatively fair results at the macro level, it masks many practical problems and dilemmas. In particular, the individual experience of fairness and the development of specific abilities of vulnerable groups are actually invisible with the extensive policies. According to the theory of human capability development, the focus of EE promotion should shift from extensiveness to fineness, from the equality of resources to the equality of human capability. In other words, circumstances dictate that the development of EE in China needs to move beyond macro control, taking into consideration the micro- and fine-grained factors, and consider the equitable development of individual capability.

At the beginning of each year, the Ministry of Education of the People's Republic of China issues the Key Points of Work of the Ministry of Education as a guide to the education work for the year. A textual analysis documents from 2012 to 2022 shows that "equity in education" is a core task every year, and "educational equity" still aims at promoting equality of educational opportunities and balanced allocation of educational resources (see Table 2).

**Table 2.** Key points of work of the Ministry of Education from 2012 to 2022 for the purpose of "educational equity".

| Serial Number | Tenet of "Educational Equity" | Year |
|:---:|:---:|:---:|
| 1 | We will promote fairness in education and effectively protect the people's right to receive education | 2012 |
| 2 | We will vigorously promote equity in education so that every child can become a useful person | 2013 |
| 3 | We will reform the way resources are allocated and vigorously promote equity in education | 2014 |
| 4 | We will vigorously promote equity in education and gradually narrow the gap between regions, urban and rural areas and schools | 2015 |
| 5 | We will uphold shared development and effectively protect the people's right to receive education | 2016 |
| 6 | We will vigorously promote equity in education and effectively narrow the gap between urban and rural areas, between regions, between schools and between groups of people | 2017 |
| 7 | We will vigorously promote equity in education and improve the public education service system | 2018 |
| 8 | We will enhance people's sense of gain from education | 2019 |
| 9 | We will vigorously promote equity in education and gradually narrow the gap between regions, urban and rural areas and between schools | 2020 |
| 10 | We will enhance people's sense of gain from education | 2021 |
| 11 | We will actively respond to the concerns of the people and ensure that the fruits of education development are more equally shared by all the people | 2022 |

Source: Key Points of Work of the Ministry of Education (2012–2022) http://www.moe.gov.cn/jyb_sjzl/moe_164/ (accessed on 25 February 2023).

Although some scholars point out that China's EE has entered a stage of "equal quality" from the old stage of "equal opportunity", in terms of practices, there are complications. These EE-related problems mainly exist in disadvantaged groups such as rural populations, populations in poverty-stricken areas, ethnic minority populations, left-behind children, and children who are living with their migrant parents. For a long time, disadvantaged groups have been the main target of EE, but they are still faced with the real problem of being deprived of capability.

Problems related to ethnic minority groups: Preferential policies for ethnic minority education in China can be divided into four categories: special policies for running schools (classes) for inland ethnic groups; special policies for senior high school entrance examination and college entrance examination; special policies for university enrollment; and special policies for ethnic preparatory classes in colleges and universities. Although these policies are aimed at enhancing the education opportunities for minority students, they are not adequate. According to an empirical survey conducted in 2013, the average length of education of China's ethnic minority population is shorter than the national average, and ethnic minority women are the group most affected by EI in particular [12]. Influenced by their social custom, China's ethnic minorities as a whole do not pay sufficient attention to education compared to the Han, and their motivation for education is not as strong. Worse still, the current teaching system is dominated by the Han culture, causing learning obstacles for ethnic minorities [13]. At the same time, the preferential education policy gives preferential treatment to all students with the "minority identity", which has caused a lot of new problems, such as "hitchhiking" by the dominant social strata in the ethnic groups and falsification of ethnic identity. In practice, therefore, the preferential education policy has become an "educational privilege" for some people and stimulated more inequality [14].

Problems related to migrant children: China's current household registration (Hukou) system creates a dual social structure, exacerbating EI problems to rural migrants working in cities. Since migrant workers are rural by Hukou registration but work in the cities, it is often difficult for their children to attend urban schools. According to statistics, the floating population (non-permanent resident in a region) in China reached 376 million in 2020, including about 200 million migrant workers [15]. Although China implements a college matriculation policy for migrant children in order to meet their needs for fair education opportunities, all localities, especially developed areas, take a "high-threshold policy", which overtly deprives migrant children of their opportunity to higher education. For example, in Beijing, the conditions for migrant children to take the college entrance examination are the following: their parents must hold a valid Beijing residence permit, residence registration card or work and residence permit, have a stable residence and occupation, and pay social insurance premiums for a certain number of years continuously; and the children must have a student status in Beijing and have been attending high school for 3 years in a row. However, even if they have the above qualifications, migrant children can only apply for the entrance examination of higher vocational schools. As a result, migrant children cannot realize the life state that they have a reason to cherish through free choice, which shows a lack of capability. This includes the lack of ability to choose further studies (such as lack of academic ability and insufficient ability to pay for education), the deprivation of local opportunities for further studies, or unequal opportunities for further studies (in fact, many cities, especially Beijing, Shanghai and Guangzhou, only offer vocational education to migrant children) [16].

Problems related to urban–rural gaps: China's administrative division has three categories: Urban Areas, County and Town Areas, and Rural Areas. This is the basis of China's regional education disparity management, macro-level resource allocation, and data collection and comparison for measuring urban–rural gaps. According to this logic and the Educational Statistics Yearbook of China [17], the urban–rural gap in education has almost been eliminated. Take the number of books and digital resources in the basic education stage as an example (see Table 3). Although Urban Areas have the most books and digital resources, followed by County and Town Areas, and then by Rural Areas, the

number of students in these three types of areas are also in the same descending order. Therefore, from a per-student perspective, there is little difference in educational resources between the three types of areas; moreover, the average number of books and computers available per student in Rural Areas is even higher than that in Urban Areas and County and Town Areas. Does this mean no regional difference in education in China? Only in numbers, not from the perspective of capabilities. This is because not everyone can turn the provided educational resources into the same or similar advantages in life [8] (p .276).

**Table 3.** China's urban–rural gaps in terms of the number of books and digital resources for basic education (2020).

| | Type | | Urban Areas | | County and Town Areas | | Rural Areas | |
|---|---|---|---|---|---|---|---|---|
| Senior high school | Number of students in school | | 12,322,698 | | 11,716,754 | | 905,077 | |
| | Number of books | Number of computers | 562,340,117 | 3,050,865 | 425,263,938 | 1,865,754 | 39,032,944 | 201,294 |
| | Quantity per student (rounded) | | 46 | 0.25 | 36 | 0.16 | 43 | 0.22 |
| Junior high school | Number of students in school | | 19,029,366 | | 23,733,472 | | 6,378,055 | |
| | Number of books | Number of computers | 663,943,679 | 3,307,564 | 866,688,808 | 3,499,747 | 286,113,228 | 1,186,617 |
| | Quantity per student (rounded) | | 35 | 0.17 | 37 | 0.15 | 45 | 0.18 |
| Primary school | Number of students in school | | 42,030,976 | | 40,717,741 | | 24,504,815 | |
| | Number of books | Number of computers | 949,166,676 | 622,984 | 933,357,062 | 301,632 | 696,694,344 | 206,818 |
| | Quantity per student (rounded) | | 39 | 0.015 | 23 | 0.0074 | 28 | 0.0084 |

For example, schools carried out large-scale online teaching due to COVID-19 in 2020. The digital teaching crisis revealed that the actual digital resources available to urban and rural students depended on the capability of students' families, rather than the number of school computers shown in the above statistics. A survey in 2020 showed that 70.62% of students in Urban Areas and County and Town Areas used computers or tablets to attend online classes, while 74.98% of students in Rural Areas used mobile phones [18] (p. 63). Therefore, it was clear that the urban–rural gap in education was further widened due to the urban–rural gap in family capability. Admittedly, educational aid can help low-SES students to a certain extent, which is also one of the important educational policies in China. In 2014, 2018, 2020 and 2021, the Chinese government provided subsidies of RMB 11.51 billion, 18.98 billion, 16.89 billion and 16.43 billion, respectively, to ordinary high school students, and most of the funds flowed to the central and western regions where low-SES students were concentrated (see Figure 1). However, it remains unclear to what extent these subsidies have improved the capability of disadvantaged groups.

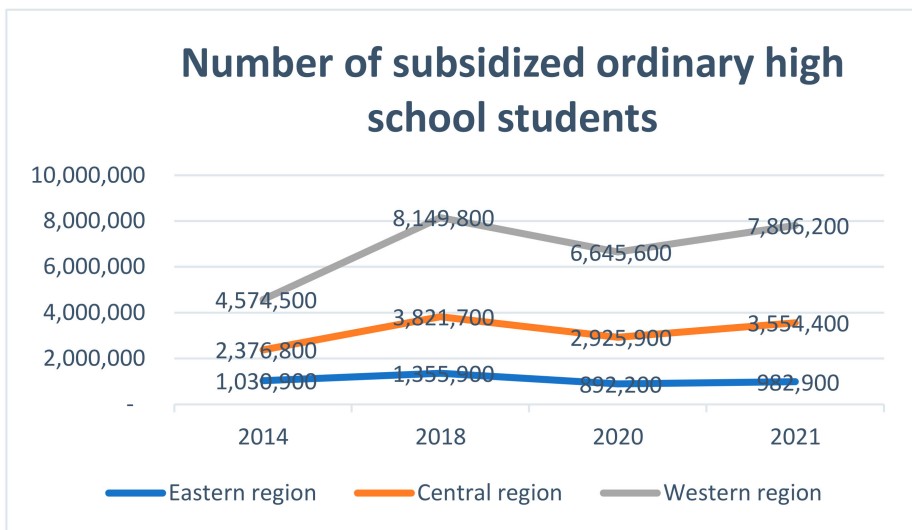

**Figure 1.** The number of subsidized ordinary high school students. Data comes from China National Center for Student Financial Aid. https://www.xszz.edu.cn/n85/index.html(accessed on 3 March 2023).

According to the latest statistics, the gross enrollment rate of higher education in China reached 57.8% in 2021, and the population with higher education backgrounds reached 240 million. MOE China declares, institutionalizing the famous Martin Trow Elite-Mass-Universal triptych on the development of Higher Education in a country, that China marking that the Chinese population now has a universal higher education. Although increasing university enrollment opportunities seems to be able to correct social injustice, it actually masks the inequality of opportunities for low-SES students to transform their capability into functions [9] (p. 70). This is because low-SES students cannot effectively convert resources into functions like their counterparts from more affluent background, and they have various obstacles to the realization of functions in school. This is more obvious after they start attending college. A "study on low-SES students' adaptability to campus life" shows that students from poor families have lower adaptability and longer adaptation period than urban students after entering the universities. This research samples students in China's 'World Class Universities' and 'World Class Disciplines', a token for their exceptional ability to get great scores in college entrance examinations. Despite their extraordinary academic performance, the researchers find that university students from low SES backgrounds are less "dedicated", less concentrated on learning, less interested in the discipline they learn, and introverted in interpersonal relations [19]. This means that even if low-SES students have the same academic achievements as high-SES students, there is still a big gap between them, which is caused, in essence, by the gap in capability.

Therefore, the equality of educational resources and results does not necessarily mean the equality of education. According to Nussbaum, among the three ways to promote social justice, the resource-based approach and the preference-based approach cannot solve the problem of inequality but aggravate it instead. In fact, the problem with fair allocation of resources is that different individuals have different capabilities to transform resources into functions. This includes both physical differences and social (hierarchical) differences [20] (pp. 232–233).

Therefore, the capability approach opposes the traditional view of fair distribution of resources and criticizes the traditional theory about what inputs (ideas, teachers and teaching materials) form specific opportunities to achieve the expected results (economic growth or social solidarity). The capability theory holds that an evaluation of social (including educational) arrangements must be based on people's capability, rather than the resources they can obtain or the results they can achieve [5] (pp. 2,4). Therefore, according to Sen's concept of equality, what should be equal is not resources (such as equitable

allocation of education funds) or results (such as the leveling of students' qualifications), but people's capability, that is, what people can become and do [5] (p. 3).

This means that we need to evaluate educational development based on people's capabilities, accurately identify the capabilities of different groups according to their economic, cultural, capital and social status, reconstruct the EE development framework based on the full investigation and experiments according to the basic idea of expanding the capability set beyond the macro allocation of resources, and reach a more complex, diverse, individualized and refined level so as to truly enhance the ability of disadvantaged groups through education.

## 3. Capability Building: Actionable Research Application on Educational Equity

The capability approach has been constantly questioned since it was put forward. One of the most famous questions is "To what extent is the capability framework operable?" [21] (p. 1953). Although Nussbaum criticized Sen for not establishing a clear framework and developing the list of ten core capabilities by means of analytical philosophy, Sen himself has always opposed setting a fixed list of capabilities, advocated an engaged human development model, and emphasized the importance of public participation and dialogue in achieving valuable capabilities. This means that some form of participatory dialogue is always needed in evaluating education-related capabilities [5] (p. 12). According to Santos Mehrotra's research, it is only at the community level that the capability approach can be really practical and useful [22] (p. 306). In this sense, actionable research is one of the effective ways to solve the EI problem using the capability approach.

Actionable research, proposed by Edward St. John, aims to identify social and educational inequities and their root causes. Through actionable research, we can reflectively choose the policies and action strategies from multiple options to address these inequities. The specific steps include identifying problems, collecting data or conducting surveys, determining solutions, taking actions, evaluating effects, and revising policies and practices, and all of these steps require cooperation among researchers, policymakers and practitioners [23] (p. 147).

The capability approach is essentially an evaluation method, so in order to make this method operable, it is necessary to determine the conditions that lead to the realization of simple functions (such as being able to read and write) and complex functions (such as being able to participate in community life and having self-esteem). Embedding the capability approach into the actionable research model means that when evaluating the topics that need to be reformed in the research-based action inquiry model (AIM), researchers and practitioners need to jointly determine the conditions for individuals to realize simple functions and complex functions; evaluate whether the development of education makes it possible to realize these functions; generate problems on this basis; conduct empirical research or data collection; and further determine the generative system to expand students' capability while following quantitative and qualitative evaluation methods. However, the measurement of capability and function is the most important challenge in applying the capability approach in empirical research and specific social environment, and the pluralism of Sen's evaluation framework actually makes research, policymaking and practice more complicated [24]. In comparison, Nussbaum's view that emphasizes the necessity of a list of core and universal human capabilities and supports the establishment of a national standard for specific capabilities is more operable as the basic way for different scholars to apply Sen's framework.

It is the challenge of building a system of capabilities and the unique role of actionable research in addressing this challenge that makes it possible to apply the capability approach to deal with educational inequalities through actionable research. That is to say, one of the important ways to construct the capability framework of disadvantaged groups is to design or create a basic capability list as a reference standard based on a large number of empirical studies and discussions under a specific social background so as to identify and examine

the capabilities that a certain group lacks and the specific factors thereof, to explore, develop and enhance the educational projects or methods for the identified capabilities.

Following the three methods of constructing the capability list commonly used in the world, Chinese scholars have studied China's capability lists of three aspects: "expert-selected capabilities", "more complicated rules and procedures for identifying capabilities" and "listening to the voices of disadvantaged groups" [6] (p. 6). The available lists present different results (see Table 4) [25] (p.80) [26] (p.122). This shows that the construction of capability lists are different when the groups they face, the problems they try to solve and the resources they have are different. In fact, this is in line with the viewpoint of the capability approach that recognizes the heterogeneity and diversity of human beings and pays attention to group differences. It is precisely because of this that capability building has become an effective way to solve the EI problem substantially. However, the available lists lack public participation and dialogue, while the capability approach emphasizes the importance of public participation in discussions and rejects paternalistic decisions. Sen clearly pointed out that decisions on what capabilities should be chosen should not be made only by local elites or cultural experts without the participation of direct stakeholders [27] (pp. 31–32).

**Table 4.** Comparison of two capability lists in China.

| Chinese Citizens' Representative Capabilities | Migrant Children's Capability of Integrating into Society | |
|---|---|---|
| Health status | Health | Physical health; mental health; social adaptation; moral integrity |
| Education | Learning | Language learning; knowledge and information learning; policy learning and compliance |
| Leisure | Social communication | Social communication |
| Income | Participation | Educational participation; community participation |
| Economic satisfaction | | |
| Trust | | |
| Free choice | | |

Actionable research advocates the comprehensive use of three capability construction methods: analytical and philosophical construction at the academic level, empirical research on social reality, and letting disadvantaged groups speak out. The three methods used in combination will help put forward a representative list of capabilities.

However, creating a capability list is only the first step. To truly expand the capability set of disadvantaged groups, collective or organized activities are needed with the list as a reference standard. In other words, solving the problem of inequality in practice also requires cooperation among local schools, communities, governments, enterprises, foundations and other parties through public- and private-funded social networks. Taking the education of migrant children in Shanghai as an example, the city has formed an effective and refined cooperative group to solve the problem of EI for migrant children.

- Educational institutions conducted special investigations and studies. In 2011, for example, the Committee for Migrant Children's Education of the Chinese Society for the Study of Tao Xingzhi carried out a "Research on the Education of Migrant Children after Junior High School in the Context of Urbanization" and published the *Blue Book of Education of Migrant Children* in Shanghai, which pointed out the education-related problems faced by migrant workers' children in Shanghai and provided an important reference for further policy formulation.

- Governments at all levels in Shanghai promulgated and implemented a series of educational policies for children of migrant workers.
- Non-governmental educational organizations hold special seminars. Since 2002, Shanghai has regularly held national special seminars on the education of migrant children so as to draw more social attention to the EE of migrant workers' children, build a communication platform and trigger further reforms.
- Support came from all walks of life. Different community groups in Shanghai participate in improving the education of migrant children. For example, the Shanghai Municipal Committee of the Communist Youth League and the Shanghai Committee of NPSC-YPC used the Children's Palace to carry out social activities for the children of migrant workers; Shanghai universities used the summer vacation to carry out the "Hand in Hand with Love" campaign, in which college student volunteers and migrant children held education activities together; Hong Kong You Dao Foundation made donations to improving the education of migrant children (see Figure 2).

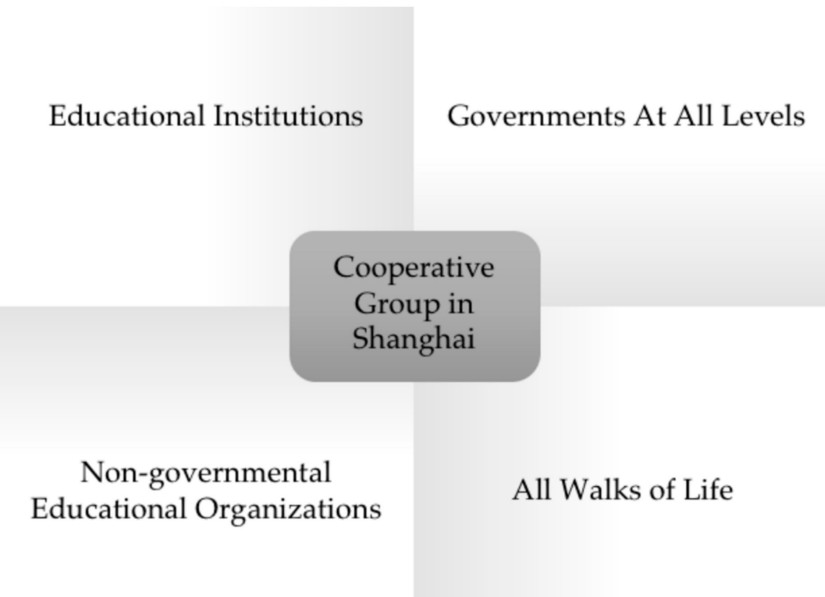

**Figure 2.** Cooperative groups in Shanghai.

Thus, the basic model of actionable research for constructing a list of capabilities for EE begins with the construction of a list of capabilities through engaged scholarship, using the functions presented by the list of capabilities as evaluation factors to locate and analyze the reasons for the lack of viable competencies of disadvantaged groups. This is followed by an analysis of the personal and social environment of disadvantaged groups and what influences them to transform what they already have into enabling resources, thus exploring ways of enhancing viable capability sets. Finally, the list of capabilities is practiced and revised through the collaboration of individuals, families, communities, and governments (see Figure 3).

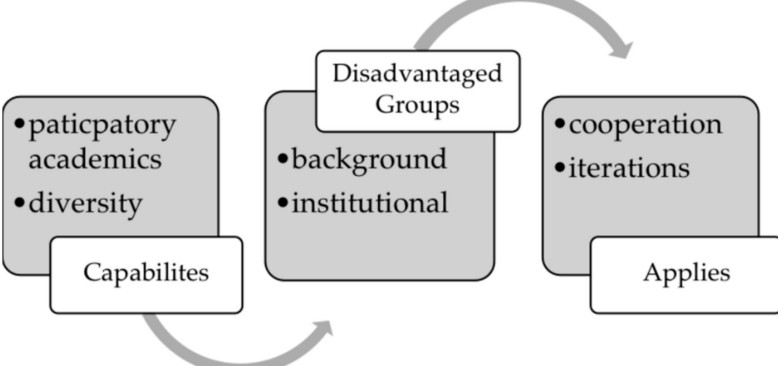

**Figure 3.** Actionable research route to build a list of capabilities.

### 4. Case Study: The LFCF Program and Its Efforts on Educational Equity

As mentioned above, actionable research is a principal and practical means to solve the EI problem through the capability approach. But what can be implemented specifically? To respond to this question, we further introduce our educational experiment in collaboration with the Chen Yidan Foundation.

In 2020, to promote the development of people's capability, we joined the Learner for the Future Competence Frame (LFCF) program launched by the Chen Yidan Foundation and established an education expert workshop. In the past two years, we have been carrying out research and experiments on how to re-endow learners with a subjective status. The core idea of the workshop is that education should focus on enhancing the development of student capabilities rather than emphasizing quantitative achievements, such as test scores. Therefore, a collaborative educational ecology involving students, families, schools and communities was built. The ultimate aim of the LFCF program is to develop learners with infinite growth possibilities, meaning that individuals develop sufficient viable competencies over the course of education and be able to achieve their own goals for a better life.

Referencing actionable research principles, the LFCF project is a collaborative educational dialogue and experiment with a rich hierarchy of participants, including students, parents, K-12 education practitioners, university academics, social activists and relevant government personnel, as well as a team of international scholars. The professors range from those with a Western cultural background to those who are native to China. The native professors are from several key universities in China, such as Tsinghua University, Peking University, Beijing Normal University and Zhejiang University. Participating K-12 education practitioners include both educators from regular schools, such as the Affiliated High School of Peking University, and educators from innovative schools, such as Avenues International School. Participating students and parents mainly come from the educational experiments conducted by the LFCF project. After numerous talks, seminars and workshops, we presented our final research report. "Competence and Education of the Learner for the Future" [28] was presented at the international education forum "Learning Ecosystem for the Future: Family, School, Society" on 12–13 November 2022. The two core contents of the report are a competence framework for the infinite growth of individuals and a PBL-dominated comprehensive practice system of educational grammar.

For students, the most critical step in capability building is the improvement of their ability, which is also one of the core goals of the LFCF project. In the early stage of the project, the LFCF team held a workshop and invited various stakeholders to discuss and analyze 763 literacy databases formed by 45 global literacy frameworks. Based on the scenario and social reality of Chinese students, a capability list (see Table 5) was preliminarily developed and a PBL-dominated community education model was proposed to effectively help students develop those capabilities. With this list as a benchmark, Chen Yidan Foundation launched the "Mars Rescue Plan" summer camp in the Shenzhen Mingde

Experimental School in July 2021. They provided a set of PBL education courses oriented to the development of the capability list with the students who entered the camping. At the end of the camping, they showcased relevant achievements in a seminar and collected comments and suggestions on this educational activity from the participating students and their parents. Then, we further adjusted and optimized the capability list based on empirical results. After two years of refinement and re-calibration, we can say that the learners in the context of China should have five basic capabilities: exploration, critical thinking, collaboration, creativity and care. These five capabilities are what people as both individuals and part of society should develop. They transcend the social structure stratification and group differences and lead to the ultimate equality of all individuals in realizing their capabilities.

**Table 5.** Capability list of the LFCF project.

| C1—Exploration | C1-1 curiosity C1-2 recognition of the problem | C1-3 courageous and resolute C1-4 concentration | C1-5 planning C1-6 trial and error |
|---|---|---|---|
| C2—Critical thinking | C2-1 reasoning C2-2 cling to the difference | C2-3 reflection C2-4 systems thinking | C2-5 resilience |
| C3—Collaboration | C3-1 listening to others C3-2 effective communication C3-3 emotion management | C3-4 tolerance C3-5 dependability C3-6 positivity | C3-7 sharing C3-8 implementation |
| C4—Creativity | C4-1 divergent thinking C4-2 traceability C4-3 crossover | C4-4 integration C4-5 break through the stereotype | C4-6 meta-cognition C4-7 self-efficacy |
| C5—Care | C5-1 empathy C5-2 diversity | C5-3 dedication C5-4 consciousness of duty | C5-5 consciousness of history and society |

Consistent with Nussbaum's view on the necessity of updating the list of basic capabilities, the LFCF team believes that these five basic capabilities are a relatively complete but open and growing framework. At the same time, we refined 31 secondary literacy indicators and provided specific references for curriculum development and teachers' teaching planning. More importantly, to increase the practicality and effectiveness of the list, the LFCF team set up a matching comprehensive practice system—Future Educational Grammar (CMYK), an interdisciplinary meta-programming system for curriculum projects corresponding to the capability list.

We not only try to provide a framework and effective tools for educational reform, but also work with individuals, families, communities, governments and other parties, emphasizing the diversified development of educational ecology. The capability approach advocates the participatory mode, and the development of EE also depends on the participation and practice of various stakeholders. At this stage, we have flexibly applied the capability list and the CMYK practice system in the education practice activities related to "protecting minority culture" and "developing the capability of students in remote rural areas". The following is a detailed description of these two EE practices activities.

As mentioned earlier, one of the obstacles to the EE of ethnic minorities in China at this stage is the over-emphasis of the Han culture in education. According to the capability approach, an important way to realize the EE of ethnic minorities is to focus on their own cultural needs. Only by taking their cultural characteristics and educational experience into consideration can they have the right and freedom to choose a suitable educational model. Therefore, in July 2022, the Chen Yidan Foundation, the main investor of the LFCF project, collaborated with Starry Night Chinese Multicultural, a youth team dedicated to protecting, inheriting and innovating multi-ethnic cultural heritage. Together, they launched a five-day project-based learning camp on Haqniq culture inheritance and innovation in the Meng Song Primary School, Mengsong Village, Menglong Town, Jinghong City, Xishuangbanna

Dai Autonomous Prefecture, Yunnan Province. The detailed itinerary of the summer camp is as follows:

1. The team went to Meng Song village to conduct research. They noticed that the majority of the residents of Meng Song Village are Haqniq. The Haqniq culture has a long history and rich content. It has unique folk customs in dance, textiles, and architecture. However, under the impact of popular culture brought about by television and the Internet, local Haqniq children have limited channels to systematically learn their own ethnic culture both at home and in school.

2. In order to further preserve the Haqniq culture, the team created a project-based learning school curriculum for the Meng Song Primary School based on the theme of "Hani Cultural Inheritance and Innovation". The curriculum is based on the LFCF capability list and the CMYK practice system and is geared towards the cultivation of the two major competencies of "inquiry" and "innovation".

3. The team conducted a five-day curriculum practice in Mengsong Primary School structured around four modules: "Cultural Insight—Cultural Inclusion—Cultural Reflection—Cultural Practice". The syllabus of the four modules is shown below (see Table 6).

4. After the summer camp, the mentors further improved the PBL school-based curriculum and teacher training materials according to the feedback from all parties involved in the teaching process, which helped the curriculum to run sustainably at the Meng Song Primary School. This means that the educational philosophy of the LFCF program which is guided by capability has taken an important step forward in realizing the EE of ethnic groups.

**Table 6.** The syllabus of the camp.

| Module | Lesson | Capability |
| --- | --- | --- |
| Cultural Insight | Haqniq's architecture | C1-1 Curiosity |
| | Haqniq's costume | |
| | Haqniq's traditional festival | |
| | The colorful world of tea | |
| Cultural Inclusion | The community of Minorities | C5-2 Diversity |
| | Interaction with Minorities from other cities | |
| | Unearthing cultural treasures | C5-4 Consciousness of duty |
| Cultural Reflection | Fieldwork around Mengsong | C1-2 Recognition of the problem |
| | Create a cultural map of Mengsong | C4-4 Integration |
| | Learn how to arrange an exhibition | C4-1 Divergent thinking |
| Cultural Practice | Arrange a Haqnia culture exhibition | C4-7 Self-efficacy |
| | Guided tours | C5-4 Consciousness of duty |

In addition to the development of EE for ethnic minorities, the capability lists and CMYK practice system are also systematically applied to disadvantaged groups in rural and remote areas and further optimized after a series of educational practices. In February 2023, the Chen Yidan Foundation cooperated with the Sustainable Education Innovation Alliance to launch the project of "promoting the development of literacy education and the realization of equitable education in rural areas". In chronological order, the operation mechanism of this project in the coming year (2023.2–2023.12) is presented (see Figure 4).

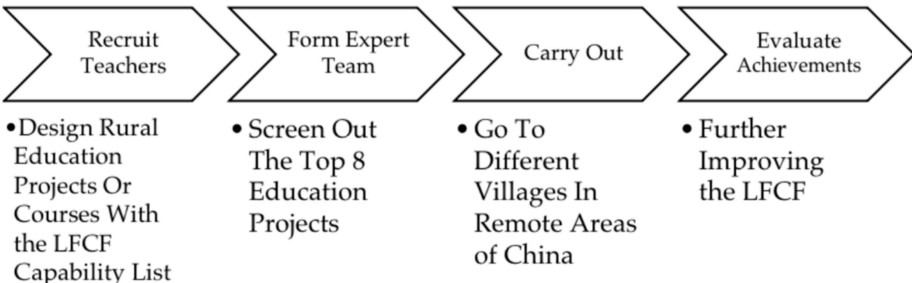

**Figure 4.** The action of "promoting the development of literacy education and the realization of equitable education in rural areas".

So far, more than 80 volunteer teaching teams have signed up to participate in this initiative, many of which aim to improve the education of disadvantaged groups. A sign language volunteer teaching team from the Xuzhou Qiyan Public Service Center conducted a sign language course on "How to make deaf students in special schools and students in general schools equally and jointly inherit the Yi cultural heritage?". Another team from Shenzhen University conducted the project "How to guide left-behind children to explore the environment and nature in their neighborhoods under the concept of sustainable development?". Although this action is still in the early stage at present, and its effectiveness needs further observation, we hope that this year-long educational action will help literacy capability education to take root, sprout and grow better in rural areas.

Based on actionable research approach, the LFCF project and its existing educational experiments bring together stakeholders to have an equitable dialogue and exchange and develop a series of educational actions to promote the development of human capability. Although officially China has multiple approaches in place to expand university enrollment and increase educational opportunities in rural areas, using engaged scholarship partnership can accelerate this process, because social justice is a commonly shared pursuit [25] (p. 60). The LFCF team's three-year action and its initial results show that engaged scholarships are of great value for and significance to the development of EE. At the same time, it also shows that human-capability-based action research about EE and social justice also needs to apply "engaged scholarship" in order to build a knowledge system that eliminates the deeply rooted and increasingly widening structural, institutional, social and cultural inequalities. As St. John and others pointed out in 2018, it is still a lofty mission of the academic community to participate in academic research that supports social actions and efforts to reduce various inequities in human society [29] (p. 51). It should be noted that the focus of the engaged scholarship is not on critique but on construction, which is also the core of actionable research. Educators should not only describe or criticize the current situation of the world, but rather think about how our actions today could create a future better educational world.

## 5. Conclusions

We advocate the combined use of the concept of capability and actionable research to organize multi-stakeholder conversations in order to deal with the problem of EI. The way we look at education is challenged by a paradigm shift from an economic orientation to human development, with human beings placed at the center of education and human capability development. A human-centered perspective is premised on expanding how people can choose to live freely, rather than promoting socio-economic development. Although some progress has been made in the development of educational equity in China at this stage, there are differences in the ability of individuals to convert resources into functions, and the government's emphasis on a balanced allocation of resources at the macro level cannot actually solve the problem of educational equity completely. From the Human Capability Perspective, we will constantly ask whether education enables individuals the "capability" to freely choose their lifestyles. We will explore possible

solutions and work together to promote equity in education. In doing so, challenges arising from the gap between theory and practice are inevitable. However, more dimensions of educational equity and more possibilities for improving educational inequalities will be demonstrated. It can be said that the LFCF project and the educational experiments that we have undertaken are a prototype of such possibilities.

Investigating and solving the EI problem from the perspective of human capability needs more voices from scholars, policymakers, social groups, and disadvantaged groups. Based on the theory of human capability, the reform of educational equity is no longer a matter of filling in or repairing, nor is it a matter of fine-tuning data measurement or stratification structures, but an innovation of individual educational development based on human capability and a holistic educational turn needs more joint actions of all social groups.

**Author Contributions:** Conceptualization, M.L. (Mingmei Li), M.L. (Min Liu) and C.W.; methodology, M.L. (Mingmei Li); figures, M.L. (Mingmei Li) and M.L. (Min Liu); writing—original draft preparation, M.L. (Mingmei Li), H.W., M.L. (Min Liu) and X.H.; writing—review and editing, M.L. (Mingmei Li) and M.L. (Min Liu); supervision, C.W.; funding acquisition, C.W. All authors have read and agreed to the published version of the manuscript.

**Funding:** This paper is funded by the International Joint Research Project "Education Development and Social Justice", Faculty of Education, Beijing Normal University (ICER201908).

**Institutional Review Board Statement:** Ethical review and approval were waived for this study as we did not collect any information that could identify the participants during the data collection process. Therefore, according to the ethical review regulations, such research does not require ethical review.

**Informed Consent Statement:** Written informed consent has been obtained from the subjects involved in the study.

**Data Availability Statement:** The data that support the findings of this study will be made available from the authors upon reasonable request.

**Acknowledgments:** The authors would like to thank Chen Yidan Foundation for supporting the actionable case.

**Conflicts of Interest:** The authors declare no conflict of interest.

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
