# Peer review of "Research on the Development of Equitable Education in China from the Human Capability Perspective"

_education, doi:10.3390/educsci13070738_

Round 1

Reviewer 1 Report (Previous Reviewer 1)

I found the paper acceptable with revision the first time and I think the revisions the authors have made improved the paper.  The issues are of critical importance.  There are still some minor editing issues that seem to come up in the revised sections of the text.  There are some word choice issues that suggest a review for English language usage may be helpful.  Figure 1 should use different color lines of the graph and the author should make reference to the figures before they are placed in the body of the text.

I think the paper is improved and is ready for publication with some copy editing and some references to the figures in the text.  The quality of the English has improved in the paper.

Author Response

Thank you for your patient work.

We have revised and re-checked the editing issues. In Figure 1, we have chosen different color lines and added the source of the data.

Reviewer 2 Report (Previous Reviewer 3)

I did review the revised version of the manuscript. The author(s) have made significant revision of the manuscript. There is an attempt on their part to lift up the theories used, the approach, the way in which data were generated and the conclusion. However, what needs to be done is to indicate the way in which data were analysed. It may also be helpful to indicate the way in which themes were developed in order to help the reader understand and realise that the conclusion is based on the data that were generated. Furthermore, there is a need for a detailed section on ethical issues especially that author(s) have  indicated that participants were interviewed. For example, negotiation for access, consent, anonymity, privacy etc. There is also a need for another editing phase of the manuscript. 

There is need to improve some of the sentences. For example, sentence no. 49, sentence no. 50, sentence no.61, sentence no 277.

Author Response

We appreciate your effort to review our manuscript and your positive feedback.

We reviewed and edited the manuscript again, correcting some editorial and grammatical issues. Your suggestions on the case study are enlightening and we have added a table showing how to use the Capability List of Table 5. In this paper, we try to use a sample to illustrate the steps and process of using “Capability” and actionable research methods to promote educational equity.

Regarding the ethical issues, we did not collect or provide any information in the private domain and did not have any direct interviews with participants. In this step, we use research feedback from another collaborative team to adapt the “Capability” tool. The team has conducted some investigations and some curricula with the “Capability” tool and their research is still ongoing. What is certain is that the tool has helped to identify some previously unnoticed Capability gaps during their investigation.

Reviewer 3 Report (New Reviewer)

The submitted manuscript is interesting in its summary, however it presents a series of aspects that could be improved that I list below, because in the current form in which it is presented from my point of view it is not publishable: SUMMARY: - I recommend redoing the summary following the introduction, objectives, method, results and conclusions sections. INTRODUCTION: - Review throughout the entire manuscript that the references comply with the journal's standards. - The introduction should end with the research questions or hypotheses that are intended to be achieved with this work. - It is not clear why do this research METHOD: - This is the weakest section of your manuscript. Directly there is no method. - When the participants are described, it would be convenient to add a table showing the main socio-demographic characteristics of the sample. - I recommend structuring this section in: or Design or sample or Instruments o Data analysis DISCUSSION AND CONCLUSIONS: - I recommend adding the discussion in the following sections: o Main findings of this work o The implications of these results in the field of study of the research o Future lines of research

Author Response

Thank you for your logical feedback.

Summary: we redo the summary following the introduction, objectives, method, results, and conclusions sections. 

Introduction: We have modified the references by referring to the journal’s published articles. As some of the cited articles may be in Chinese journals due to the research on educational equity being in the Chinese context, which has caused some confusion about the references. the suggestion of “The introduction end with the research questions or hypotheses” is helpful to clarify our research objective, so we try to propose the question in the introduction. The revision is highlighted in the paper.  

Method: We propose an actionable research approach combined with the human capability approach to improving multi-stakeholder collaboration in educational equity reform; and present a sample to illustrate the steps and process. we have enriched the content of the sample this time.

Discussion and conclusions: We modified the discussion in the following sections: The main perspective of this work, the implications of the sample, and the future lines of research.

Reviewer 4 Report (New Reviewer)

Below are some considerations for improvements:

Abstract: needs some adjustments: methodology used, main results found, final considerations and suggestions for future studies and study limitations.

It is suggested to start with the introduction, approaching the theme in breadth until the theme object of the study.

In the theoretical framework, it is suggested to carry out a bibliometric research on the subject, for example on the Web of Science, and write about the state of the art.

The study methodology does not contain. However, you need to show the reader what kind of study you are doing.

Nothing to declare

Author Response

Thank you for your inspiring comments.

We did try to make some improvements to the abstract, the introduction, and the method.

Abstract: we made some adjustments to the methodology used, main results found, final considerations and suggestions for future studies, and study limitations.

Introduction: we try to separate this section and approach the theme step by step.

Methodology: the information is provided in the abstract of this paper.

Round 2

Reviewer 3 Report (New Reviewer)

congratulations

This manuscript is a resubmission of an earlier submission. The following is a list of the peer review reports and author responses from that submission.

Round 1

Reviewer 1 Report

I found this to be a generally well written scholarly paper building upon the human capabilities model that St. John and others have developed over time.  I also found the paper gets stronger after the introduction so the only concerns I have for the paper are with the initial framing.

Generally, I found the introduction to be confusing.  The authors jump into some complex concepts well before they define them in the text.  I would like to see a clearer statement regarding the significance of employing the human capabilities approach in China as a way to address issues of educational inequality.  For example, the authors deal with a very important policy issue later in the paper regarding the education of migrant children moving from Western rural communities to the Eastern urban centers.  This would be a great way to introduce the paper as well because it is a significant issue with real implications for society.  Starting abstract and remaining abstract well through the first half of the paper leaves me as the reader wondering why this is significant.

I would also suggest that if the authors are going to deal with big ideas in the introduction like Marx's theory of economic capital or Bourdieu's conception of social and cultural capital, they need to provide the reader with a bit more articulation of those ideas.  I am not sure the audience for the journal, but I found myself needing more information to assess the quality of the case they were intending to make.

After the introduction, I find that the purpose of the paper becomes clearer and it is written more clearly and more authoritatively.  It is a strong paper overall, with a need for a stronger introduction at the front end.

I do find some editing issues that I will address in the next section, possibly related to English language, but there is a specific term they use several times that suggests it is not a spelling error.  They refer to "n-tuple" twice starting on page 3 and then separately refer to "tuple" after that.  I have no idea what that means and the others do not introduce it in any formal way.

I think the paper could use some minor editing for English language proficiency.  For the most part it is well written, but there are some grammatical errors occasionally throughout the document that suggest that portions of the paper were written by non-native English speakers.  For example, missing an occasional "a" or "the" to allow text to flow more smoothly.  I think this may be a copy editing issue that can be resolved easily.

Reviewer 2 Report

This article reviews different studies on the empowerment of low socio-economic students as a way to promote more equitable education. However, this review lacks systematisation, falling far short, in my opinion, of the rigour that should drive a scientific article. Therefore, I have doubts about the choice of articles, the criteria for inclusion and exclusion of sources, the objective of the article, the method used, etcetera. No guidelines accepted in the academic world are provided to ensure rigour in the conduct of the review (Example: http://prisma-statement.org/?AspxAutoDetectCookieSupport=1). 

Reviewer 3 Report

The manuscript is fairly well edited. It is conceptually strong and it focuses on very interesting isssue of Human Capital and Human development in educational reform. It is an extended review of substantive research studies and it analyses issues of educational reform which has an international resonance. Furthermore, it has the potential to extend the readers' theoretical understanding of approaches to educational reform where China is used as a case study.

However, there are few editorial issues that I noted. For example, the use of a capital letter after a comma in page 1, line 22. It is not an empirical paper, consequently, the author(s) should declare that this is not an empirical paper, but it has used secondary sources to craft the paper. Furthermore, the use of actionable research needs further clarification. When reading the paper, I thought that the author(s) are intending to reflect on the way in which they used actionable research to change and improve the situation. However, I later realised that they were reviewing the use of actionable research that was done by other researchers. Threre is therefore a need to explain that this a review of the way in which actionable research was used by other researchers in educational reform.  There is a need to justify the focus on actionable research. The way it is written, also affect the cohesion of the manuscript. Lack of cohesion is also reflected on the conclusion that starts in page 15 line 588. For example, the conclusion is on the shift from economy-oriented to human-oriented and less on actionable research. There is also a need to justify the selction of China for use in the manscript. It may help to enhance the context of the paper. What also needs to be highlighted is the purpose of the paper. That will help the reader to read with the purpose in mind.

The quality of English used is high.